# DynamiX: Dynamic Resource eXploration for Personalized Ad-Recommendations

**Adam Holeman** *   **Sohini Roychowdhury** *   **Mohammad Amin**   **Feng Wei**
**Bhaskar Mehta**   **Srihari Reddy**
Meta, Ads Data and Representation learning

## Abstract

For online ad-recommendation systems, processing complete user-ad-engagement histories is both computationally intensive and noise-prone. We introduce *Dynamix*, a scalable, personalized sequence exploration framework that optimizes event history processing using maximum relevance principles and self-supervised learning through Event Based Features (EBFs). *Dynamix* categorizes users-engagements at session and surface-levels by leveraging correlations between dwell-times and ad-conversion events. This enables targeted, event-level feature removal and selective feature boosting for certain user-segments, thereby yielding training and inference efficiency wins without sacrificing engaging ad-prediction accuracy. While, dynamic resource removal increases training and inference throughput by 1.15% and 1.8%, respectively, dynamic feature boosting provides 0.033 NE gains while boosting inference QPS by 4.2% over baseline models. These results demonstrate that *Dynamix* achieves significant cost efficiency and performance improvements in online user-sequence based recommendation models. Self-supervised user-segmentation and resource exploration can further boost complex feature selection strategies while optimizing for workflow and compute resources.

## 1   Introduction

Sequences of user activity are powerful sources of user-behavior and task understanding, which have demonstrated significant benefits across various Ads-recommendation systems [8]. In prior works [8] [6] [12], event-based-feature (EBF) sequences are described to combine data from multi-modal sources gathered from user-engagement activities or events (such as content viewing, likes etc.) into organic sequences that capture dynamic user interests. Examples of EBF sequences include user-ad-impressions, organic feed impressions, video_view etc., wherein time-stamped-ordered sequences are represented by a vector of features about a specific *event*. Ad-recommendation systems with input EBF sequences, while capable in accurately gauging dynamic user interests, incur high storage and compute complexities to ensure relevance and freshness for all real-time traffic [7]. Additionally, user-ad engagement sequences have unreliable labels thereby leading to noisy low-intent data streams, that can lead to unpredictable user engagement levels. This necessitates investigations into elastic resource exploration and allocation strategies, with the goal of eliminating training and serving capacities for less-engaged users while boosting the capacity for better engaged users, respectively.

Recent work on user-segmentation in [9] [10] demonstrate that certain user demographics benefit more from historic-usage data sources when compared to other users with only marginal impact. This implies that by truncating the length of the input EBF-sequences for users who do not benefit from a given data source, we can create personalized EBF sequences which maintain model accuracy while reducing overall compute costs. We refer to this approach as static resource allocation since it relies on long-term usage-patterns for user-segmentation. In this work, we consider an alternative approach: on-the-fly or dynamic feature exploration at run-time. This approach filters the inputs to

---

*Equal contribution

the model during the training stage so that the model only receives a subset of the truncated-features in the forward pass [12], thereby decreasing the costs of training and inference, respectively. The self-supervision process can explore across different EBFs in real-time based on the user's current session, through on-the-fly computations and select the optimal EBF-sequences per user per-engagement event. This implies that the same user can get different events dropped at different times of the day owing to the self-supervised sequence filtering strategy. This work builds upon the existing active area of the *Dynamic feature selection* research in [1] [11] [4], especially, for domains where the most informative predictors can vary based on context.

ML-based feature selection approaches [13] [2] typically compare the input features to the training labels to only include features that are highly predictive. However, dynamic feature exploration must estimate the discriminative power of inputs using only the information available at inference-time. We address this problem by using the past conversion behavior of users as a replacement for positive training labels to perform feature exploration and subsequent feature selection, in the form of self-supervision, thereby relocating compute resources from passive users to engaged or active users. Figure 1, demonstrates this variation between static and dynamic resource allocation processes, wherein existing static user-segmentation works in [10][5] rely on user-segments based on pre-assigned user-importance, while the proposed dynamic user-segmentation enables users, per-surface, to be categorized into active vs. passive classes at different times of the day based on usage. Based on this user-segmentation, low-priority feature attributes can be removed from

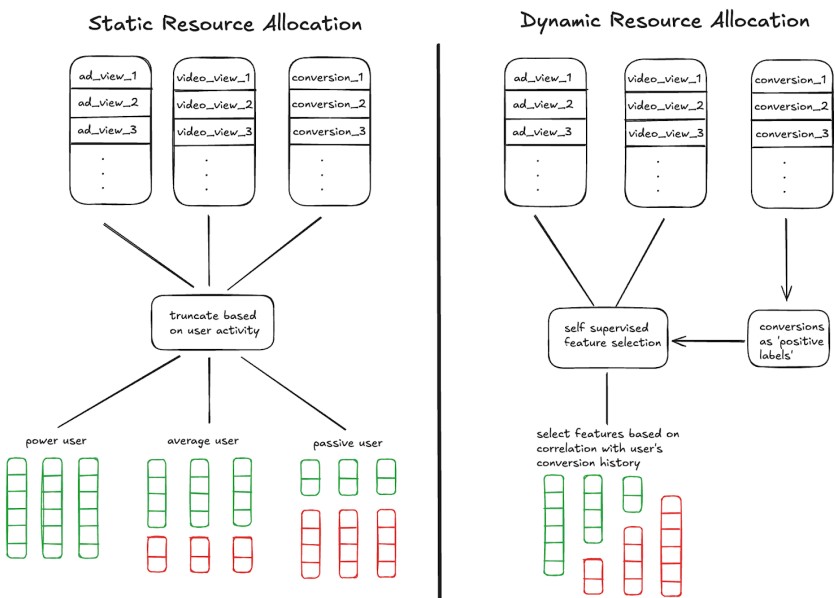

Figure 1: System diagrams for personalized-sequence scaling by static and dynamic feature selection methods. Predictability of user-content engagement remains relatively stable for passive-users while active or engaged users can be represented as power-users and averagely engaged users.

active-users and high priority features can be boosted for active-users to secure training and inference efficiencies and click-through-rate (CTR) prediction gains. There are three noteworthy properties of EBFs that enable such dynamic resource exploration [7]. First, EBFs capture most informative ephemeral features per-user. Second, content-engagement signals that are provided by an input EBF can also be estimated by comparing the feature to a user's conversion history. Third, certain passive user-groups are less sensitive to feature freshness and inference-time compute resources with regards to aggregated CTR.

This paper makes two major contributions. First, we present a novel self-supervised approach for detecting correlations between a user's ad-view history and their conversion history. This enables run-time segmentation of users into active and passive groups through a collection of pre-processing transformations. Second, we propose *Dynamix*, that is a maximum relevance approach that dynamically selects EBFs conditional on the runtime-allocated user-groups, thereby enabling selective feature-attribute removal, feature selection and boosting for personalized ad-recommendation systems.

## 2  Prior Work

Related works on user-segmentation for real-time data traffic are shown in Table 1. Most of these works rely on generating user-ad embeddings followed by unsupervised clustering methods to generate user-segments and subsequent boosting for specific user/content groups. In this work, we

Table 1: Summary of User-segmentation methods for Real-time Traffic Data Sources.

| Method | Method | Data Used | Contributions | Results |
|---|---|---|---|---|
| Yang et al., 2013 [5] | Hidden Markov models to improve dynamic user-behavioral segments. | Web browsing logs | Modeled temporal evolution of user-profiles for adaptive ad-targeting. | 13-15% improvement over random in page prediction accuracy. |
| Zhou et al., 2018 [13] | Embedding-based ad personalization framework. | Ad click/conversion logs | Demonstrated improved targeting via deep user representations. | Deep L0 Ranking: +0.032% NDCG; Semantic Candidate Generation: +0.37% in-segment Ads score; Graph-based retrieval: +1.78% ads score overall |
| Cui et al., 2019 [2] | Advanced segmentation with deep-learning and attention mechanisms. | Multi-source user-data | Utilized attention/embedding for fine-grained, personalized targeting. | Pre-ranked ads caching: +0.07% to +0.19% ads score impact; IG Reels PAE trigger expansion: +0.03% ads score; Personalized AdIndexer eCPM filtering: -0.02% ads score overall. |
| Victorator et al. 2021 [10] | Unsupervised segmentation (k-means). | Browsing/ad interaction logs | Enabled scalable, data-driven discovery of user-groups. | Clustering improved offline training NE by õ.07%; Cluster-based Embedding Retrieval for FBE Ads: +4.9% Ads Score. |

propose a novel dynamically time-varying self-supervised user segmentation approach that enables resource exploration and feature-selection for the segmented user-groups.

## 3  Dynamix for Personalized Ad-recommendation

Traditionally, ad-recommendation systems are trained to predict user engagement levels per content-type based on historic trends of personalized usage. However, the training and inference resources for all users per-surface personas are kept consistent [6]. In this work, we introduce a novel dynamic resource extrapolation method called *Dynamix*, which is a *maximum relevance approach* to dynamically remove EBF features and sources for certain user-segments and dynamically selecting EBFs for active user-groups to boost overall ad-relevance and user-ad engagement.

Given user-engagement sequences in the form of EBFs, *Dynamix* involves user-segmentation to active and passive categories across small time windows followed by selective feature shrinkage and boosting for the different user-segments, respectively. The goal is to increase the training and inference resource ROI and boosting aggregated CTR predictions by providing more engaging content to active user-segments and getting passive user-segments to be more predicable. In the following sub-sections, we define the impact of dwell-time, which is the time a user spends viewing a piece of ad or content, towards user-conversion events, which are represented by clicks or any other engagement/conversion events (e.g. like, share, comment etc.).

### 3.1  User-engagement with Dwell-Time Attribute

In this section, we connect the dwell-time on ad-impressions EBF to conversion events. Our approach applies a user's conversion history to extract training labels for the supervised classification problem, and then applies the user dwell-times as the predictive variables [3]. We begin with the following notation for each user $u$:

- $D_t^u$ - the dwell-time of an ad-impression at time $t$.
- $C_t^u$ - the timestamps of past conversion events. $C_t = 0$ for no-conversion event and $C_t = 1$ for a click or conversion event.

For EBF sequences [7], the timestamps form a discrete set and due to logging discrepancies the timestamp of an ad-impression may differ from the timestamp of an associated conversion event. Thus, it is useful to extend the definition of $C_t^u$ to windowed time-intervals to allow comparison of impressions and conversions as shown in Figure 2.

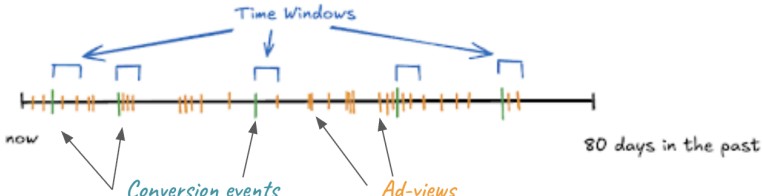

Figure 2: Example of time-windows applied to compare the average dwell-time of conversion events within windows to that outside the windows to capture statistically significant trends in a user's dwell-time behavior.

Given a time-window $s$, we define $C_{[t,t+s]}^u$ per-user as (1).

$$C_{[t,t+s]}^u = \begin{cases} 0 & \text{if there is no conversion in the interval } [t, t+s], \\ 1 & \text{if there is a conversion in the interval } [t, t+s], \end{cases} \tag{1}$$

where, the parameter $s$ as the *forecast horizon*. Here, we make the simplifying assumption that the conditional random variables are normally distributed as (2).

$$log(D_t^u)\big|C_{[t,t+s]}^u \sim \mathcal{N}(\mu_C^u, \sigma_C^u), \tag{2}$$

where, the parameters of the distribution depend not only on the user, but also the binary variable $C_{[t,t+s]}^u$. Empirical evidence of this distribution can be observed in Figure 3. Here, we observe that for many users there are significant differences in the distributions $log(D_t^u)\big|(C_{[t,t+s]}^u = 1)$ and $log(D_t^u)\big|(C_{[t,t+s]}^u = 0)$.

Next, we apply the Bayes' rule to the probability of a conversion-event given a significant dwell-time in (3)-(6).

$$p(C_{[t,t+s]=1}^u\big|log(D_t^u)) = \frac{p(log(D_t^u)\big|C_{[t,t+s]}^u = 1)p(C_{[t,t+s]}^u = 1)}{p(log(D_t^u))}, \tag{3}$$

$$= \frac{p(log(D_t^u)\big|C_{[t,t+s]}^u = 1)p(C_{[t,t+s]}^u = 1)}{p(log(D_t^u\big|C_{[t,t+s]}^u = 1))p(C_{[t,t+s]}^u = 1) + p(C_{[t,t+s]}^u = 0))p(C_{[t,t+s]}^u = 0)}, \tag{4}$$

$$= \frac{1}{1 + \frac{p(log(D_t^u)|C_{[t,t+s]}^u=0)p(C_{[t,t+s]}^u=0)}{p(log(D_t^u)|C_{[t,t+s]}^u=1)p(C_{[t,t+s]}^u=1)}}, \tag{5}$$

$$= \sigma(-w \cdot log(D_t^u) + b). \tag{6}$$

Here, $\sigma$ is the logistic sigmoid function and $w$ and $b$ are functions of the following 5 distribution parameters: $\mu_{C_t=1}^u, \mu_{C_t=0}^u, \sigma_{C_t=1}^u, \sigma_{C_t=0}^u$, and, the prior probability of the user converting on a window of size $s$, or $P(C_{[t,t+s]}^u)$, which, we assume to be independent of $t$. For simplicity, we also assume that $(\sigma_{C_t=0}^u = \sigma_{C_t=1}^u)$ for all users $u$.

The historical values of the user's conversions and dwell-times $\mathcal{D}^u = (D_t^u, C_{[t,t+s]}^u)$ allow us to fit these parameters to the user's history using a maximum-likelihood estimate in (7).

$$w_{MLE} = \text{argmax}_w p(\mathcal{D}^u) = \frac{\mathbb{E}[log(D_t)\big|C_{[t,t+s]} = 1] - \mathbb{E}[log(D_t)\big|C_{[t,t+s]} = 0]}{S^u}, \tag{7}$$

where, $S^u$ is a variance estimator and the expectation ($\mathbb{E}$) represents the empirical average over the observed data.

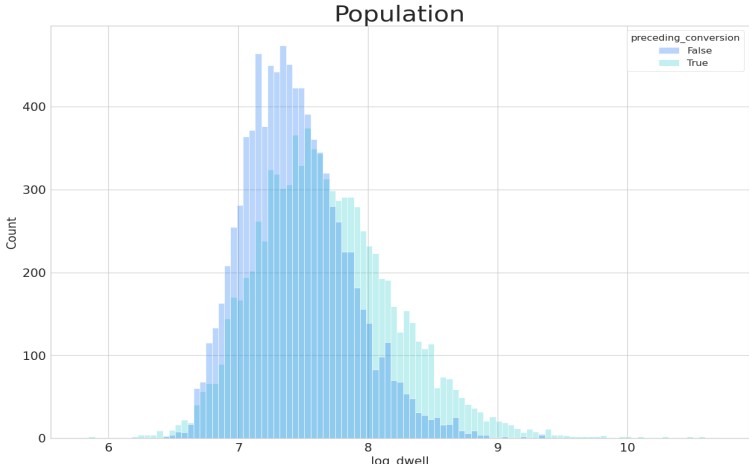

Figure 3: Histogram of log dwell-times in the 5 minutes preceding a conversion event (green) vs. more than 5 minutes before the next conversion event (blue).

It is noteworthy that shifts in dwell-time before conversion events demonstrate the correlation between ad-history and conversion history. In Figure 3, we observe that on an average, dwell-time increases before conversion events. This implies that users spend more time looking at ads in the few minutes prior to engaging with an advertisement. This distribution also demonstrates that for a 5 minute window prior to a user-ad-engagement event, users typically tend to engage with multiple ads.

Additionally, we note that different users exhibit different degrees of correlation and the same user can demonstrate positive, low and negative correlations at different times of the day as shown by 3 sample users in Figure 4 below. We utilize this dynamic correlation to implement user-segmentation and subsequent preferential resource exploration.

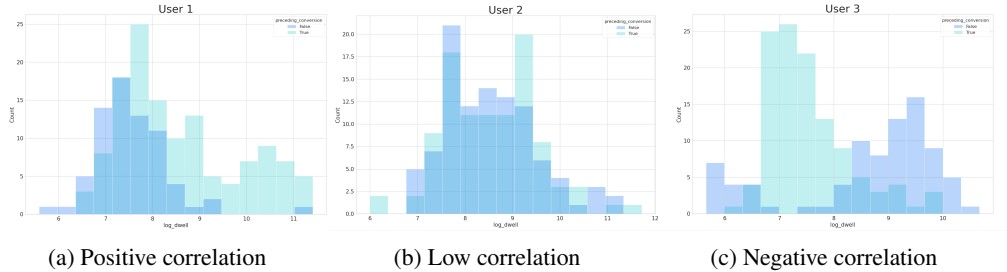

(a) Positive correlation       (b) Low correlation       (c) Negative correlation

Figure 4: Different dwell-time patterns across individual users

## 3.2 Dynamic User-segmentation and Selective Feature Exploration

We utilize the normalized maximum-likelihood estimate in (7) to evaluate the windowed correlation statistic between dwell-time and conversion events per-user in (8).

$$\text{Corr}_s^u(D, C) := \mathbb{E}[\log(D_t)|C_{[t,t+s]} = 1] - \mathbb{E}[\log(D_t)|C_{[t,t+s]} = 0]. \tag{8}$$

Here, $\text{Corr}_s^u(D, C)$ compares the user's average dwell-time on advertisements they viewed *close in time* to conversion events and their average dwell-time on advertisements *far away* from conversion events, where proximity is controlled by the forecast-horizon $s$. It is noteworthy that on an average, $\text{Corr}_s^u(D, C) > 0$, as depicted in Figure 3, has lower variance when compared to the correlation distributions for individual users in Figure 4, thereby demonstrating the noisy nature of labels at user-level.

Next, the windowed absolute correlations statistic is thresholded to determine the passive and active users in (9)

$$U_s^u = \begin{cases} 0 & \text{if } |\text{Corr}_s^u(D, C)| <= \epsilon, \\ 1 & \text{if } |\text{Corr}_s^u(D, C)| > \epsilon \end{cases} \qquad (9)$$

The dynamic threshold-parameter $\epsilon$ is empirically computed to separate the most positively and negatively correlated uses (or active-users) from the uncorrelated, or passive-users. This difference in engagement statistic is intuitive since we observe that users within the passive-segment $U_s^u = 0$ are those with noisy ad-view history, while users within the active segment $U_s^u = 1$ are those whose ad-view history is highly predictive of commercial intent. Since the user-segments can update multiple times a day for individual users based on their browsing history (from Figure 4), pre-processing functions are implemented in conjunction with the EBF sequence at the source, to facilitate *Dynamix*.

Finally, we assess the improvement in CTR for user-ad recommendations in terms of Normalized Entropy (NE), which defined as follows:

$$\text{NE Gain} = -\frac{\frac{1}{N} \sum \left( y_j \log p_j + (1 - y_j) \log(1 - p_j) \right)}{\hat{p} \log \hat{p} + (1 - \hat{p}) \log (1 - \hat{p})}, \qquad (10)$$

where, $y_j$ are actual labels, $p_j$ are model predictions, $\hat{p} = \frac{\sum y_j}{N}$ is the prior probability and $N$ is the total number of user-samples. The storage and compute resource savings are evaluated in terms of positive Queries-per-second (QPS) gains for model training and inference, respectively.

### 3.3 Production Data

The definitions of EBFs used for the dynamic user-segmentation and feature selection are defined as follows.

- Organic-impression EBFs: This event is generated when user-generated content (UGC) is displayed on the user-screen (website or mobile-app) with $>= 50\%$ of the content being visible, and the user views for at least 250 milliseconds. Event attributes include {content-id (unique identifier), dwell-time (duration of visibility), media-type (e.g., image, text, video), position (rank in the user's personalized feed), Timestamp}.

- Ad-impression EBFs: This event is generated when an ad-content is displayed with greater than 50% visibility and user views for at least 250 milliseconds. Event attributes include {semantic-ids (content understanding model-generated metadata), ad-id, Timestamp}

- New-page impression EBF: This event is generated when users click or engage with specific pages. Event attributes include {semantic-ids (content understanding model-generated metadata), media-type, Timestamp}

All experiments are trained on over 40 billion user-samples curated over a month of usage in one-pass to demonstrate the NE gain curves.

Additionally, there were two major production-data level challenges in computing $\text{Corr}_s^u(D, C)$. First, the conversion events and impression events are logged through different systems and we often observe that the impression event corresponding to a given conversion event is logged up to a minute *after* the conversion event is logged. Second, many impression events are logged with unrealistic dwell-times, extending from a few milliseconds to several hours.

To mitigate the first challenge, we added a logging buffer in our computation of the correlation statistic (8), replacing $C_{[t,t+s]}$ with $C_{[t-60,t+s]}$ to account for logging delays in the ad-impression EBF. Further, we mitigated the second challenge by de-noising the ad-impression events based on dwell-time to remove outliers before computing $\text{Corr}_s^u(D, C)$.

## 4 Experiments

To assess the importance of user-group specific resource-level exploration, we performed two sets of experiments. In the first experiment, we selectively remove low-importance feature attributes for active and passive user groups with the goal of improving training and inference QPS. In the second

experiment, we add EBF feature-attributes selectively for active and passive user-groups, with the goal is enhancing overall user-ad engagements. This selective EBF attribute removal and boosting demonstrates the impact of *Dynamix* for personalized sequence-based ad-recommendations.

## 4.1 Dynamic Feature Removal for Compute Wins

In this experiment, we applied dynamic feature removal from baseline EBF sources. The production baseline contained 11-attributes corresponding to the ad-impression EBF-data source and we applied dynamic feature selection to 4 of these attributes. These 4 attributes were selected based on low feature importance-ranking and they have only incremental impact on the ad-recommendation model performance. For this experiment, two evaluation runs applied dynamic feature removal to the passive and active user segments, respectively, to reduce training and inference feature costs as shown in Table 2. Empirically, 66.67% of the ad-traffic users were segmented at any given time as active-users while remaining 33.33% were segmented as passive-users. The empirical threshold-parameter $\epsilon$ was selected to maximize the NE gains between two user-segments. Further, we observed that both runs resulted in small NE Gains, but both yielded training QPS wins that were proportional to the traffic reductions.

Table 2: Averaged metrics for EBF-attribute removal experiments to user-segments across 3 runs.

| Experiment: For Ad-impression EBF | NE Gain | Training QPS | Inference QPS |
|---|---|---|---|
| Attribute removal for passive users, No change for active users | 0.012 | +0.4% | -0.37% |
| Attribute removal for active users, No change for passive users | **0.006** | **+1.1%** | **+1.8%** |

It is noteworthy that NE Gains in the range $[0, 0.02]$ and $QPS < 1\%$ can be considered as experimental noise. From this experiment, we conclude that pruning less significant feature attributes from active user-segments result in minimal change in averaged-CTR across all users while significantly speeding up training and inference processes as shown in Figure 5(a). The observation for training and inference QPS gains for active user-segment only is explainable. The passive user-group suffers from low-correlation between conversion-events and dwell-time, thereby such user-ad engagements are less predictable even after feature pruning when compared to the active user-group. Conversely, since the active user-group is more predictable for ad-intent, pruning low-importance feature attributes results in consistent efficiency for training and inference compute.

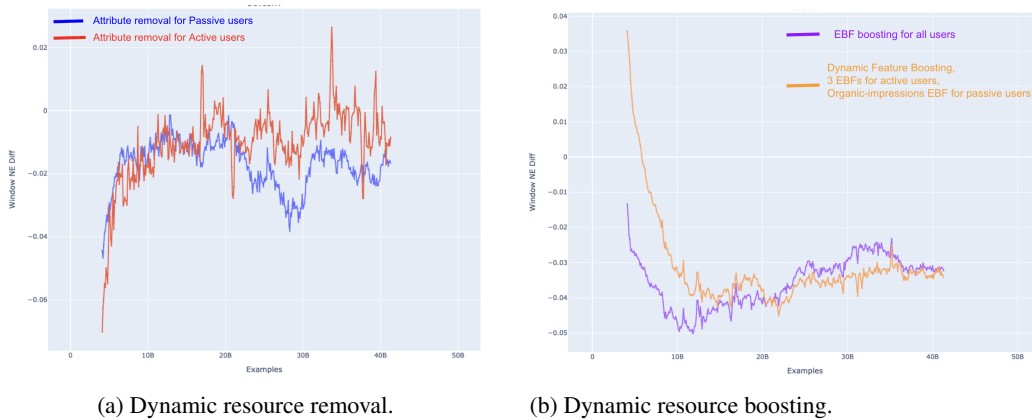

(a) Dynamic resource removal.  (b) Dynamic resource boosting.

Figure 5: Convergence patterns for Dynamic resource removal and boosting, thereby leading to dynamic feature selection.

## 4.2 Dynamic Feature Boosting for Ad-recommendation Gains

In this experiment, we applied dynamic feature selection to selectively add feature-attributes to 3 EBF sequences corresponding to ad-impressions, organic-impressions and new-page-impressions.

*Dynamix* involved user-traffic-segmentation using ad-impressions EBF into 66.67% active and 33.33% passive-users (with same $\epsilon$ as Section 4.1), respectively. In the first run, feature-attribute additions for all 3-EBF sources were applied to all users homogeneously. In the second run, the attribute additions to all 3 EBF sources were made only to the active user-segment while the passive user-segment received additional organic-impressions feature only.

Table 3 demonstrates that user-segmentation followed by selective feature boosting can not only enhance training and inference QPS by traffic reduction, it can subsequently boost overall CTR prediction through stable NE Gains shown in Figure 5(b).

Table 3: Averaged metrics for selective resource addition across 3 runs.

| Experiment: New Attribute Addition to | NE Gain | Training QPS | Inference QPS |
|---|---|---|---|
| 3 EBF sources for all users | 0.031 | -2.13% | -10.1% |
| 3 EBF sources for active-users, Organic-impression EBF for passive-users | **0.033** | **-0.96%** | **+4.2%** |

It is noteworthy in Table 3 that while the impact to training QPS is minor ($\approx 1\%$ QPS), dynamic user-segmentation and feature boosting leads to 14.3% inference QPS improvement over uniform EBF boosting across all users. This is a significant improvement over the homogeneous feature addition run for all users that leads to greater than 10% inference QPS regression, and hence is compute heavy and infeasible.

## 5 Conclusion and Discussion

In this work, we presented *Dynamix*, a personalized sequence scaling approach that dynamically allocates resources based on user-engagement patterns in user-ad interactions. Our experiments demonstrate the effectiveness of user-group specific resource-level exploration through selective feature removal, feature selection and attribute-boosting. Dynamic feature-attribute removal showed that pruning low-importance attributes for active users significantly improved training and inference throughput (up to +1.1% and +1.8% QPS respectively) without degrading model-prediction. This confirms that targeted feature pruning for predictable user-intent can reduce computational costs while maintaining recommendation quality.

Conversely, the dynamic feature boosting revealed that selectively adding feature attributes to active-users, while limiting organic feed features only for passive users, led to a substantial 4.2% increase in inference QPS and a significant NE Gain. This personalized boosting approach outperformed uniform feature addition, which caused significant inference regressions, highlighting the importance of user-segmentation for efficient resource allocation.

It is noteworthy that the dynamic user-segmentation process is sensitive to the threshold parameter thereby enabling multi-user group segmentation by multi-level thresholding. The binary user-group segmentation presented in this work is the simplest extension to the baseline EBF-based ad-recommendation model to demonstrate ad-recommendation boosts while maintaining stable processing wins. Additional user-group partitioning may lead to more noisy and unstable resource explorations that remain to explored in future works.

Overall, *Dynamix* enables cost-effective and performance-enhancing personalized sequence scaling by leveraging self-supervised user engagement signals. Future work will focus on refining correlation estimation methods, multi-user-segments while maintaining optimizing and explainable workflows to further improve computational efficiency and recommendation quality.

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
