# OpenReview forum: "DynamiX: Dynamic Resource eXploration for Personalized Ad-Recommendations"
_NeurIPS.cc/2025/Workshop/Reliable_ML — NeurIPS 2025 - Reliable ML Workshop_

### Official Review · Reviewer_piNk · 2025-09-15
**Promising Direction with Useful Insights, Yet Preliminary in Evidence**

**Rating:** 6
**Confidence:** 3

**Review:**

### **Summary**

The paper introduces Dynamix, a framework for dynamically managing feature resources in large-scale ad recommendation. The key idea is to prune input features on the fly: for “passive” users, low-value event-based features are dropped to improve efficiency, while for “active” users, important features are retained or even boosted to enhance accuracy. The classification of users is driven by self-supervised dwell time–conversion correlations, updated several times a day. Experiments on production traffic show consistent QPS improvements and modest but meaningful NE gains, demonstrating that pruning and selective boosting can reduce computational cost without sacrificing performance.

### **Strengths**
* Novelty: Applies dynamic feature pruning and boosting at the input level for large-scale recommendation, a practical yet under-explored approach.
* Empirical quality: Demonstrates consistent efficiency gains (QPS) with modest accuracy improvements on production-scale data.
* Clarity: The intuition (using dwell time as a self-supervised proxy for intent) is straightforward and well explained.
* Relevance: Align with the workshop’s theme of robustness under imperfect data.

### **Weakness**
* Missing comparisons: No baselines against state-of-the-art dynamic feature selection or conditional computation methods; limited ablation on which features are pruned/boosted.
* Proof gaps: No formal guarantee that pruning preserves accuracy beyond empirical validation; NE gains lack confidence intervals.
* Failure modes: Risk of reinforcement feedback loops where passive users are persistently under-served; no exploration of safeguards.
* Scope limits: Results are aggregate; little analysis of performance for cold-start users, rare events, or shifts in campaigns.

### **Suggestions for Authors**
* Include stronger baselines.
* Provide statistical significance and sensitivity studies on thresholds and dwell-time assumptions. For example, explain why NE gain between [0. 0.2] is considered experimental noise.
* Add per-segment analysis (e.g., device, geography, cold-start) to demonstrate robustness.
* Consider exploration or fairness mechanisms to prevent passive users from being systematically pruned.
* Release anonymized data samples and code (or at least reproducible simulation scripts) to support replicability of results.

### **Ethics**
* Privacy: Clarify how per-user dwell times and conversions are stored, anonymized, and retained.
* Fairness: Ensure that pruning does not disproportionately disadvantage certain user groups.
* Misuse: Dwell-time optimization could incentivize manipulative designs (e.g., clickbait); safeguards should be considered.

---

### Official Review · Reviewer_PLor · 2025-09-17
**Overall positive review**

**Rating:** 6
**Confidence:** 3

**Review:**

Summary
This paper proposes Dynamix, a scalable and personalized sequence exploration framework designed for improving ad-recommendation systems. The central idea behind Dynamix is to perform on-the-fly, or dynamic, feature exploration at run time. This is coupled with a novel self-supervised approach that detects correlations between users’ ad-view histories and their conversion histories. Based on these correlations, users are segmented into two categories: active and passive. The segmentation then informs downstream steps such as selective feature-attribute removal, feature selection, and boosting. Together, these mechanisms enable the recommendation system to adapt in real time, providing a more personalized and efficient experience for users. The authors conduct a set of experiments demonstrating that this framework yields consistent improvements in performance, suggesting its practical value for real-world large-scale recommendation settings.

Strengths
The main strength of the paper lies in the novelty of its methodology. Although feature selection and user segmentation are not new topics in the recommender-systems literature, combining these two ideas into a unified and scalable framework is original and impactful. The introduction of dynamic feature exploration at run time is particularly compelling, as it addresses the limitations of static feature sets that may not adapt well to changing user behavior. Similarly, the binary segmentation into active and passive groups provides a simple yet effective mechanism to personalize recommendations based on user engagement levels. Another notable strength is the scale of the experiments: the authors conduct evaluations on large-scale data, which lends credibility and reliability to the reported improvements. This large-scale empirical validation suggests that Dynamix is not merely an academic concept but has real potential for deployment in industry contexts where efficiency and scalability are critical. Overall, the contributions are both technically interesting and practically relevant.

Weaknesses and Limitations
While the contributions are strong, the framework is not without limitations. As the authors themselves note, the binary segmentation into only active and passive categories may be overly simplistic. User behavior is often more nuanced and may fall into a spectrum of engagement levels rather than a dichotomy. Extending the segmentation to multiple classes or continuous measures of activity could potentially capture richer patterns of user interaction and further improve recommendation quality.

Suggestions for the Authors
One concrete suggestion is to conduct another round of experiments where the segmentation threshold parameter, epsilon, is systematically varied. Such an analysis would provide insight into the sensitivity of the framework to this design choice and help identify whether more fine-grained segmentation could be beneficial. In addition, exploring multi-class segmentation or adaptive thresholds could represent a valuable extension to the current binary approach.

Ethics
The paper does not raise any notable ethical concerns. The focus is on methodological innovation and empirical evaluation within the scope of recommendation systems. No new ethical risks are introduced beyond those already associated with personalization and recommender technology in general.